# STIM-Orai Channels and Reactive Oxygen Species in the Tumor Microenvironment

**DOI:** 10.3390/cancers11040457

**Published:** 2019-03-30

**Authors:** Janina Frisch, Adrian Angenendt, Markus Hoth, Leticia Prates Roma, Annette Lis

**Affiliations:** 1Department of Biophysics, Center for Integrative Physiology and Molecular Medicine, Medical Faculty, Saarland University, 66421 Homburg, Germany; Janina.Frisch@uks.eu (J.F.); Adrian.Angenendt@uks.eu (A.A.); markus.hoth@uks.eu (M.H.); 2Center for Human and Molecular Biology, Saarland University, 66421 Homburg, Germany

**Keywords:** Orai, STIM, calcium, reactive oxygen species, H_2_O_2_, tumor microenvironment

## Abstract

The tumor microenvironment (TME) is shaped by cancer and noncancerous cells, the extracellular matrix, soluble factors, and blood vessels. Interactions between the cells, matrix, soluble factors, and blood vessels generate this complex heterogeneous microenvironment. The TME may be metabolically beneficial or unbeneficial for tumor growth, it may favor or not favor a productive immune response against tumor cells, or it may even favor conditions suited to hijacking the immune system for benefitting tumor growth. Soluble factors relevant for TME include oxygen, reactive oxygen species (ROS), ATP, Ca^2+^, H^+^, growth factors, or cytokines. Ca^2+^ plays a prominent role in the TME because its concentration is directly linked to cancer cell proliferation, apoptosis, or migration but also to immune cell function. Stromal-interaction molecules (STIM)-activated Orai channels are major Ca^2+^ entry channels in cancer cells and immune cells, they are upregulated in many tumors, and they are strongly regulated by ROS. Thus, STIM and Orai are interesting candidates to regulate cancer cell fate in the TME. In this review, we summarize the current knowledge about the function of ROS and STIM/Orai in cancer cells; discuss their interdependencies; and propose new hypotheses how TME, ROS, and Orai channels influence each other.

## 1. Introduction

The tumor microenvironment (TME) (Figure 1) has a significant influence on carcinogenesis (tumor development). The TME is generated by cancer and noncancerous cells, including immune cells, cell–cell interactions, the extracellular matrix, and soluble factors. Soluble factors include oxygen; nutrients; reactive oxygen species (ROS); reactive nitrogen species (RNS); ATP; Ca^2+^, H^+^, and other ions; growth factors; chemokines; cytokines; or waste products [1,2,3,4]. The intracellular Ca^2+^ concentration ((Ca^2+^)_int_) is a key regulator of (cancer) cell proliferation and apoptosis and, thus, should play an important role in tumor growth and development. Ca^2+^ influx across the plasma membrane is a major mechanism to shaping (Ca^2+^)_int_ in all cells, including cancer and immune cells [5,6,7,8,9]. Stromal-interaction molecules (STIM)-activated Orai channels represent the main Ca^2+^ channel type in most electrically unexcitable cells including immune cells [6,7,9] but also many cancer cells [5,10,11]. Their expression in cancer cells is found to be correlated with metastatic progression, a poor prognosis, and a shorter survival. Since malignant cells exhibit a strong dependence on Ca^2+^ flux for proliferation, Orai channels could be considered a potential therapeutic target to inhibit cancer growth.

ROS have recently been in the focus of TME research because, depending on their concentrations, ROS may be decisive for the life and death of cancer cells [12,13]. Since Orai1 and Orai2 but not Orai3 channels are strongly regulated by ROS [14,15,16], Orai channels are interesting targets to integrate Ca^2+^ influx and ROS signaling in the TME. In this review, we focus on the interactions of Orai channels and ROS in the TME and on their potential relevance for TME development. We propose a scenario where redox changes alter Orai function and Ca^2+^ influx in both malignant and nonmalignant cells, such as immune cells, resulting in changes in (Ca^2+^)_int_ with a direct impact on tumor fate.

## 2. The Tumor Microenvironment (TME)

According to the World Health Organization (WHO), cancer is “the second leading cause of death globally and is estimated to account for 9.6 million deaths in 2018” (World Health Organization). The process of cancer development and progression is called carcinogenesis and is divided into 3 to 4 distinct steps called initiation, promotion, progression, and metastasis [17].

In solid tumors, the tumor mass is formed by a diverse milieu which is composed of malignant and nonmalignant cells such as endothelial cells, cancer-associated fibroblasts, immune cells, adipose cells, and neuroendocrine cells in addition to vascular and lymphatic networks and the extracellular matrix (ECM) [1]. This dynamic and complex multicellular environment is known as the tumor microenvironment (TME) (Figure 1). The TME has long been considered an important factor for tumor growth: The first publications are from the 19th century [18]! In the past few years, the TME and noncancerous cells have been recognized as major players for tumor growth [19,20] and, therefore, as potential targets for drug actions. However, due to its complexity, the TME and its cancer-type specific features still remain an obstacle for efficient cancer therapy [3]. Many of the molecular mechanisms of signaling pathways within the TME are not well-understood and the complex interactions between cellular and non-cellular TME components are not well-defined. A detailed understanding of the TME and its interactions would allow for better pharmacological treatment and a better tumor prognosis. 

Depending on the cellular and non-cellular composition, the local milieu can be highly variable between different tumors or even within the same tumor (Figure 1). The metabolism of different cell types, cell–cell interactions, the architecture of TME formed by remodeling of ECM proteins (which create a stiff fibrotic matrix), and the blood supply together create an environment composed of oxygen; nutrients; reactive oxygen species (ROS); reactive nitrogen species (RNS); ATP; Ca^2+^, H^+^, and other ions; growth factors; chemokines; cytokines; or waste products (Figure 1). This environment, together with locally secreted molecules from different cells, leads to pH gradients, differences in oxygen tension, and interstitial pressure across the tumor [2,4,21]. These parameters have an impact on the metabolism of surrounding cells and consequently cellular function, where the environment is constantly changing and adapting according to new challenges and interactions with the host. 

It has been shown that the TME can promote cancer growth and metastasis as a result of bidirectional interactions between cancer cells, noncancerous cells, and the surrounding environment. In the worst scenario, the TME forms a niche favorable to tumor growth and less favorable to other cells that could eliminate or limit tumor cell proliferation. In fact, it has been shown that most of the nonmalignant cells within the TME often adopt a tumor-promoting phenotype after the local milieu modifies their cellular functions [22]. Such alterations include changes in gene expression and cellular activity. For example, hypoxia, which develops at the beginning of tumor growth and induces cell necrosis (Figure 1), leads to the activation of hypoxia-responsive genes in malignant and nonmalignant cells. It also promotes the recruitment and survival of immune cells that are mainly glycolytic, such as macrophages, which produce large amounts of ROS [3]. This has drastic consequences for the cells present in this niche, as they need to adapt to be able to survive in this highly oxidative environment. Increased ROS also activates pathways in leukocytes to secrete more cytokines that favor tumor growth, leads to new cellular mutations, and may, therefore, transform other cells and induce apoptosis. Regions that undergo cycling hypoxia are also present in the TME. Cycling here refers to situations of fluctuating oxygen levels, going from deep hypoxia to moderate hypoxia states, similar to the reoxygenation of blood vessels in the human heart after ischemia, which bares the risk of ischemia-reperfusion injury [23]. Cycling hypoxia can affect cells near blood vessels, which are not efficiently perfused, contrary to hypoxic regions which are normally far from blood vessels [24]. Cells exposed to cycling hypoxia need to deal with and to adapt to two conditions: a lack of oxygen and sudden reoxygenation, which leads to very high amounts of ROS. All these changes can create different sub-microenvironments within the TME, each of them having distinct characteristics, populated by different cells, and exposed to different molecules.

## 3. STIM/Orai Channels in Cancer

Like all cells, cancer cells of the TME express a whole range of different ion channels. Considering the importance of Ca^2+^ for cell signaling including proliferation, migration, and apoptosis in combination with the strong dependence of cancer growth on these mechanisms, Ca^2+^ channels should play a very important role in the TME. Among Ca^2+^ channels, Orai channels have a prominent role in many electrically unexcitable cells [6,7,9]. They represent the main Ca^2+^ channel in most immune cells in which they were initially discovered [25], but they are also highly expressed in cancer cells and correlate with metastatic progression, a poor prognosis, and a shorter survival [5,6,7,8,10,11]. Since malignant cell functions depend on Ca^2+^ flux, considerable interest has emerged in the therapeutic potential of inhibiting Orai for many cancer types. 

### 3.1. Short Introduction to STIM and Orai

Store-operated Ca^2+^ entry (SOCE) is the major Ca^2+^ entry pathway including cancer and immune cells [5,6,7,8,10,11]. Its well-known activation mechanism is depicted in Figure 2. The two known SOCE activators, STIM1 and STIM2, sense the ER Ca^2+^ store content and activate Orai channels in the plasma membrane. STIMs are known to form homomultimers (as depicted in Figure 2) but may also form heteromultimers [26]. STIM1 is less sensitive to ER luminal Ca^2+^ compared to STIM2 [27] but activates Orai channels significantly stronger [28]. There are three known isoforms of the tetraspanning hexameric Orai channels, Orai1, Orai2, and Orai3 (Figure 2), with characteristic properties [29,30]. All of them are activated after the depletion of Ca^2+^ from the ER, possess a high selectivity for Ca^2+^, are inwardly rectifying, and show a Ca^2+^-dependent inactivation (CDI) [30]. 

### 3.2. STIM/Orai in Tumor Initiation and Promotion

Initiation is the first step in cancer development where normal cells undergo irreversible changes, transform, escape the immune surveillance, undergo continuous unregulated proliferation, and are able to form tumors at the end. The loss of growth control involves a whole range of critical mutations and is the sum of the accumulated abnormalities in a cell’s regulatory systems. Whether changes in (Ca^2+^)_int_ are relevant for cancer initiation is currently not clear [5].

A mechanistic link between Ca^2+^ homeostasis and chromosome instability was recently proposed in hepatocytes with Hepatitis B viral (HBV) infection, as the driver of hepatocellular carcinoma [31]. The hepatocytes carry a gain-of-function mutation in the preS2 region of a large surface antigen (LBHS), one of two HBV-encoded oncoproteins, that is linked to the early onset of hepatocellular carcinoma. The preS2 mutation promotes ER-plasma membrane (PM) connections through ER stress and causes the permanent activation of SOCE in these cells, inducing chromosome instability, aneuploidy, and anchorage-independent growth [31]. However, more in vivo studies and direct evidence are necessary to prove or disprove a role of SOCE and, thus, Ca^2+^ as cancer initiators or drivers.

### 3.3. STIM/Orai in Tumor Proliferation/Growth

Orais and STIMs are expressed in the vast majority of tumors [5,10]. Their expression levels seem to correlate with metastatic progression, a poor prognosis, and a shorter survival in studies using patient specimens. Colorectal cancer patients with a positive expression of STIM1 [32] or/and with a high Orai1 expression had poorer prognoses and shorter overall survival rates [33]. Comparable results for Orai1 were also shown for non-small cell lung cancer [34], esophageal squamous cell carcinoma [35], and gastric cancer [36]. Additionally, Orai3 expression was increased in tumor tissues of lung adenocarcinoma [37], prostate cancer [38], and breast cancer [39] and correlated with overall survival and metastasis-free survival [37]. Interestingly, an analysis of a microarray from McAndrew and colleagues revealed a significantly poorer prognosis [40] for breast cancer patients with a STIM1-high and STIM2-low phenotype. Considering that not only the ratio between STIM1, STIM2, Orai1, Orai2, and Orai3 but also the discovery of STIM2 splice variants [41,42] are highly relevant for Ca^2+^ channel activity [43,44,45,46,47], the relative composition of STIMs and Orais needs to be carefully addressed not only in tumors.

Cell proliferation is dependent on the cell cycle and transitions between different phases (G0/G1, S phase, and G2/M phase) which are tightly controlled through Ca^2+^-dependent checkpoints (reviewed in Reference [48]). SOCE alters cancer cell proliferation in vitro [49,50,51] and also in vivo [33,35,36,52,53,54]. However, how Ca^2+^ controls distinct checkpoints is not well-understood. Increases in the basal or transient fluctuation of Ca^2+^ are involved, but also (Ca^2+^)_ext_ needs to be considered. Cell cycle arrest in the G0/G1 phase in U251 cells [53], in neck squamous cell carcinoma cell lines [54], and at the S and G2/M phases in cervical cancer cells [52] by STIM1-silencing has been reported. A pro-proliferative role of STIM1 in vivo using U251 human glioma xenograft model in mice revealed that knocking down STIM1 in xenografts demonstrated a diminished growth [53]. In contrast, an elevated Orai1 and/or STIM1 expression can promote cell proliferation [36,54]. In non-small lung cancer cells, nicotine promotes cell proliferation by upregulating Orai1 expression and therefore by enhancing SOCE and increasing basal Ca^2+^ concentration [55]. In esophageal squamous cell carcinoma zinc is able to inhibit Orai1-mediated SOCE, Ca^2+^ oscillations, and subsequent cell proliferation [56]. The pharmacological inhibition or knocking down of Orai channel could block human esophageal squamous cell carcinoma proliferation in vitro and tumor growth in vivo [35]. Another study shows that high Ca^2+^ diet in a mouse model of slowly evolving prostate cancer accelerated its progression by promoting proliferation [57] indicating the importance of (Ca^2+^)_ext_. Growth factors as fibroblast growth factor 4 (FGF4) may also have an impact on Orai1 expression, resulting in increased SOCE, and may promote epithelial-mesenchymal transition and enhanced cell proliferation [58]. One of the most important and interesting cascades in this context is the mTOR (mechanistic target of rapamycin) pathway (reviewed in Reference [59]). As the catalytic subunit of two distinct protein complexes (mTORC1 and C2), this serine/threonine protein kinase plays a central role for the cell perception of the environment in the regulation of metabolism, cell cycle, and growth. It is, thus, essential for the adaptability of cells to specific changes and needs as in the TME. A recent study showed an interesting relationship between mTORC1 and STIM1 expression as a novel potential therapeutic approach for patients with tuberous sclerosis complex (TSC) tumors [60].

Not only the main players of SOCE, STIM1 and Orai1, but also the slightly “neglected” STIM2 and Orai3 are involved in proliferation and growth of tumor cells [61]. The overexpression of STIM2 inhibits cell proliferation and tumor growth in colorectal cancers in vivo [62] but promotes cell migration in primary melanoma in vivo [63], implicating the contribution of STIM2 signaling at different stages of tumor progression. Furthermore, the high Orai1 and STIM2 expression found in melanoma biopsies at the rim of invading tumors are linking their possible role in tumor invasion and/or metastasis in vivo [63]. In human carcinoma cells versus normal mucosa cells, STIM2 protein was nearly depleted in contrast to an upregulation of STIM1 and all three Orai proteins [64]. The involvement of Orai3 in the machinery of tumorigenesis has been reported, including breast, prostate, and lung cancer [38,65,66,67]. By using mice xenograft models, it was shown that Orai3 plays a crucial role in prostate cancer development in vivo [38]. The knockdown of Orai3 significantly reduced SOCE and inhibited proliferation by arresting non-small cell lung cancer cell lines in the G0/G1 phase [67]. Orai3 transcripts are differentially expressed in the different subtypes of breast cancer and regulated by estrogen receptor alpha (ERα). The silencing of ERα causes a decreased expression of Orai3 and cell proliferation in vitro [65]. The same study places Orai3 as an important player in tumorigenesis in vivo, since the growth of breast tumors was significantly reduced by Orai3 knockdown before the transfer to the recipient mice with severe combined immunodeficiencies (SCID) [65]. Orai3 expression seems to be regulated positively and negatively by miRNA and to act directly on Orai3 3′UTR [68]. Another interesting finding places Orai1–Orai3 channel complexes in the center of attention in a variety of prostate cancer cells [38]. The study reports an oncogenic switch in which cells, especially those exposed to tumor microenvironmental factors (here, arachidonic acid), change from homologous Orai1 complexes to more heterogeneous Orai1–Orai3 complexes by upregulating Orai3 expression. This switch could be an excellent adaptation, where a shift from a pro-apoptotic to more pro-proliferative phenotype is beneficial for cell proliferation and growth [38].

### 3.4. STIM/Orai in Tumor Survival/Apoptosis

A well-controlled balance between cell proliferation and cell death is necessary to avoid excessive proliferation leading to cancer development. In cancer, a scenario with too little apoptosis dominates, causing the expansion of malignant cells that are resistant to apoptosis. Despite being part of the problem, apoptosis is a popular target in cancer treatment. Therefore, a better understanding of the complex underlying mechanism of apoptosis is the key to developing more specific targets to execute the lethal hit against cancer cells. Ca^2+^ plays a pivotal role in the mechanistic induction of apoptosis. During apoptosis, (Ca^2+^)_int_ is dramatically increased, and as a consequence, mitochondria take up large amounts of it and induce apoptosis. Interrupting the prolonged Ca^2+^ influx through SOCE by knocking down Orai/Stim or blocking it with specific inhibitors can counteract cell apoptosis.

Human colon carcinoma cells show increased Orai and STIM1 expression, but STIM2 is almost depleted [64]. In noncancerous cells, the knockdown of STIM2 decreases SOCE while it promotes apoptosis resistance. This finding suggests that the loss of STIM2 contributes to apoptosis resistance in tumor cells. In addition, the blockage of STIM1-mediated SOCE can significantly enhance chemotherapy-induced apoptosis in lung and pancreatic cancer cells [69,70]. In a pancreatic adenocarcinoma cell line, a siRNA-mediated knockdown of Orai1 and/or STIM1 increases apoptosis induced by chemotherapy drugs 5-fluorouracil or gemcitabine [70]. In addition, Orai1 downregulation has been shown to contribute to the formation of an apoptosis-resistant phenotype in prostate cancer cells [71]. Furthermore, the presence of an increased Orai3 expression, leading to the assembly of more Orai1/Orai3 channel complexes can increase the resistance to apoptosis, as has been suggested in the context of pancreas carcinoma [38]. Similar results are found in breast cancer cells where the downregulation of Orai3 arrests cell-cycle progression and induces apoptosis but not in normal breast epithelial cells [39].

A recent study placed Bcl-2 as a SOCE regulator to modify ER stress-induced apoptosis [72]. The Bcl-2 family plays a major role in the regulation of apoptosis. Its pro- or anti-apoptotic members act mainly at the mitochondria level. Bcl-2 is the first identified anti-apoptotic protein capable of preventing apoptosis in a Ca^2+^-dependent manner [73]. A mutant of Bcl-2 used in the study increased the expression of SOCE components and depleted Ca^2+^ in the ER lumen, causing a massive Ca^2+^ influx leading to caspase activation and apoptosis [72]. Additionally, several anticancer drugs that are used to induce cancer cell apoptosis function through the dysregulation of Ca^2+^ signaling, for example, in colon cancer cells or triple-negative breast cancer [74,75].

### 3.5. STIM/Orai in Epithelial-to-Mesenchymal Transition (EMT)/Cancer Progression

Changes in cell phenotypes, defined as epithelial-to-mesenchymal transition (EMT), are important for tumor metastasis [76,77]. This transition is associated with an improvement in migratory and invasive properties. The list of EMT inducers is long, including growth factors secreted by the tumor environment, cytokines, hypoxia, and metabolic changes. Additionally, the indispensable change in gene expression is activated by complex regulatory networks, involving transcriptional control, transcriptional factors, miRNAs, alternative splicing, posttranslational regulation, protein stability, and subcellular localization [78]. Several studies already reported that an altered SOCE is linked to EMT in prostate cancer [79], colon cancer [32], and gastric cancer [36]. Transforming growth factor β1 (TGF-β1) in MCF7 breast cancer cells enhanced SOCE. Silencing the transcription factor Oct4 or significant inhibition via TGF-β1 upregulated the expression of STIM1 and Orai1 and promoted invasion and metastasis by inducing EMT [80]. TGF-β-induced EMT seems to be differently regulated by the expression of STIM2 (regulating non-store-operated Ca^2+^ entry) and by the expression of STIM1 (regulating store-dependent Ca^2+^ entry) [81]. Another study describes the Orai3 and STIM1 requirement for TGF-β-dependent Snai1 transcription, a transcription factor upregulated during EMT [82]. A similar study in lung adenocarcinoma cell lines shows that FGF4 was also able to induce EMT by elevating Ca^2+^ entry via the expression of Orai1 channels [58].

### 3.6. STIM/Orai in Tumor Metastasis/Angiogenesis

Cell motility is partly mediated by a (Ca^2+^)_int_ gradient [83,84], and several components of migration mechanisms, such as cytoskeleton remodeling, leading edge guidance, and matrix degradation, are Ca^2+^ sensitive [85]. Over the last ten years, in vitro and in vivo evidence has accumulated that SOCE components are involved in cell motility, invasion, and tumor metastasis [86]. The inhibition of SOCE or their components inhibit metastasis of breast cancer [57,87,88], melanoma [63,89], colorectal cancer [32], prostate cancer [51], and gastric cancer [36]. Consistently, the overexpression of STIM1 enhances cell migration in cervical cancer [52] and colorectal cancer [49]. Cell motility and, thus, metastasis are regulated by dynamic interactions between cytoskeleton, myosin II and focal adhesions [90], which assemble and disassemble to mediate cell migration [91]. STIM1-dependent signaling regulates focal adhesion turnover [52,87] required for leading edge protrusion and trailing tail retraction and regulates actomyosin contractility [92]. Furthermore, silencing STIM1 significantly alters podosome dynamics, reduces cell invasiveness [93], and regulates the dephosphorylation of focal adhesion kinase (FAK) by modulating focal adhesion turnover [94,95] and the recruitment and association of active pTyr397-FAK and talin at focal adhesions [92]. Similar results were found in breast cancer cells in a murine tumor metastasis model [87], in colorectal cancer following a destabilization of STIM1 [96], and in glioma cell lines [95].

One important pathway to modulate cell migration, invasiveness, and metastasis is the PI3K/Akt/mTOR signaling pathway. Akt plays a central role by phosphorylating many proteins involved in the stabilization of actin cytoskeleton and by promoting migration via remodeling. SOCE is positively regulated by the PI3K/Akt pathway, and this effect might be suppressed by targeting receptor tyrosine kinases (RTK) [97]. The RTK protein family includes epidermal growth factor receptors (EGFRs), FGF receptors, and vascular endothelial growth factor receptors (VEGFRs) [98] which are heavily involved in cell migration [98] and angiogenesis. Upregulated VEGF production by a high STIM1 expression in human cervical cancer cells regulates the focal-adhesion dynamics of migratory cells [52]. Accordingly, the inactivation of PI3K/Akt signaling pathway by STIM1 knockdown reduced the migration and invasion of prostate cancer cells [51]. Furthermore, a lipid raft SK3/TRPC1/Orai1 complex promotes cell migration in metastatic colorectal cancer. The formation of this complex is favored by the phosphorylation of STIM by epidermal growth factor (EGF) and the activation of Akt [99]. A SK3-Orai1 complex also plays a critical role in cell migration and bone metastasis [88,99].

The proliferation and motility of cells are critical steps in angiogenesis. Due to continuous and fast growth, tumors rely on the formation of new blood vessels to ensure an adequate supply of oxygen and nutrients [100]. Low oxygen tension (hypoxia) promotes metastasis and is considered as the key driver in angiogenesis [101,102]. Hypoxia is sensed by the tissue and triggers the cellular production of hypoxia-inducible factor 1 (HIF-1), a transcription factor that activates many downstream pathways [103] including the expression of VEGF, TGF-β, and platelet-derived growth factor (PDGF-β) [104]. Hypoxia is a common feature of the TME of most solid tumors, and hypoxic cancer cells secret VEGFs to initiate tumor angiogenesis [100,105]. Interestingly, VEGF and FGF (and others) increase (Ca^2+^)_int_. However, molecular mechanisms underlying SOCE-mediated angiogenesis remain poorly understood.

In colon cancer [106] and in triple-negative breast cancer [107], hypoxia leads to the upregulation of Orai1 by the Notch1 pathway. These data are in line with the findings that Notch1 signaling pathways activate NFκB [108] and that NFκB regulates the expression of Orai1 and STIM1 [109]. Orai1 upregulation potentiates SOCE and activates the nuclear factor of activated T cells, NFAT4, contributing to hypoxia-induced invasion and angiogenesis [106,107]. Hypoxia leads to the accumulation of hypoxia-inducible factor 1-alpha (HIF-1α), a subunit of the transcription factor HIF, which responds to alterations of available oxygen [110]. The hypoxia-induced accumulation of HIF-1α was found to correlate with the overexpression of STIM1 in human and murine hepatocarcinoma cells (HCCs) [111]. The increase in STIM1 expression is a consequence of a direct HIF-1α binding to the promoter of STIM1 and leads to an increase in SOCE in HCCs promoting tumor growth [111]. During hypoxia, HIF-1α is stabilized and induces many genes like VEGF for a better adaptation to this condition [112]. Since the production of ROS is augmented under hypoxic conditions [113] and ROS inhibits the SOCE mediated by STIM1 and Orai1 or Orai2 [14,15,16], the overexpression of STIM1 might be a countermeasure of the HCCs to provide the necessary Ca^2+^ signals for vital cell functions. Furthermore, the expression of STIM2 was also reported [114] under hypoxic conditions. In rat pulmonary arterial smooth muscle cells, hypoxia-related STIM2 overexpression is accompanied by an increased SOCE and proliferation [114]. Apart from the hypoxia-induced induction of Orai1 and STIMs, it was recently also reported that Orai3 expression was induced by HIF1α in MDA-MB-468 ERα negative cells [115]. Unlike in ERα positive MCF-7 breast cancer cells [65,66] where Orai3 silencing reduces SOCE, the upregulation of Orai3 did not contribute to SOCE in ERα negative cells [115]. However, since the Orai3 expression in breast cancer is highly dependent on the ERα expression itself, one might speculate that the Orai1 complexes contribute significantly to Ca^2+^ signals in ERα negative cells. These results place the channels and sensors as new targets in regulating hypoxia.

## 4. ROS Production and Elimination

As mentioned before, reactive oxygen species (ROS) are one of the key factors to influence the TME, and they also modulate Orai channels (see below). In the following section, we summarize the most common pathways and mechanisms in ROS metabolism with specific regard to the TME inspired by two reviews [116,117].

ROS comprise a group of molecules that are generated via the partial reduction of O_2_ and establish high chemical reactivity [116,117,118]. The one-electron reduction of oxygen leads first to the formation of superoxide anion radical (^•^O_2_¯), which in turn is dismutated and further reduced to hydrogen peroxide (H_2_O_2_) which is finally either fully reduced to water or partially reduced and split to hydroxyl radical (OH^•^) and hydroxyl anion (OH¯) [119]. ROS is mainly produced by mitochondria, NADPH oxidases (NOX), and other enzymes like xanthine oxidase and cytochrome P450 [116,117,120,121] (Figure 3).

Very high ROS levels can be produced in mitochondria when ^•^O_2_¯ molecules are released during ATP generation in the electron transport chain (ETC) [116,122]. Several endogenous and exogenous factors influence mitochondrial ROS production, including mitochondrial membrane potential [123], hypoxia, and nutrient metabolites but also cancer- and immune-related factors like TNF-α and Toll-like receptors [117,124,125,126,127].

NOX complexes also produce very high levels of ROS (Figure 3). The production of ROS via the NOX family occurs during the catalysis of electron transfer from NADPH to O_2_ and thereby produces ^•^O_2_¯ [117]. NOX-dependent ROS production requires the functional assembly of the NOX complex, which can be mediated by different signaling molecules such as growth factors and, again, TNF-α and Toll-like receptors [117,128,129]. Similar to the mitochondrial ROS production, the release of ^•^O_2_¯ is dependent on the location of the NOX molecules. NOX comprises NOX1-5 and Duox1/2 [116] which are present in the plasma membrane and intracellular membranes of the nucleus, mitochondria and the endoplasmatic reticulum (ER) [117]. The specific isoforms at particular sites can lead to ROS release either to the intracellular or extracellular spaces [117,121].

ROS have long been considered to be solely deleterious for cells causing oxidative damage in different molecules like DNA, lipids, or proteins. However, moderate ROS levels are also important for various physiological cellular functions, including intracellular signaling, cell survival, proliferation and immune responses [117,122,130,131]. Hence, a unique redox homeostasis is required to control the balance between production and elimination [117]. ROS elimination, or “antioxidant defense”, is mainly performed by four enzymatic systems: superoxide dismutases (SODs), peroxiredoxin (PRX)/thioredoxin (TRX) system, glutathione peroxidase (GPX)/glutathione (GSH) system, and catalase (Figure 3).

The quantification of ROS levels is a challenging task. Table 1 summarizes the most commonly used approaches. 

For more comprehensive overviews over certain technologies, we acknowledge the following reviews which are also relevant for the conceptual design of Table 1: references [132,133,134,135,136,137,138,139,140,141,142,143,144,145,146,147,148,149,150].

## 5. Impact of Reactive Oxygen Species (ROS) in the Tumor Microenvironment (TME)

ROS likely plays a dual role in cancer, both as initiating factors as well as downstream signaling molecules as a result of malignant transformations. In the literature, numerous reviews are available that deal with cancer and ROS, but it is often hard to distinguish between cancer development and progression and between “good” and “bad” ROS, although these differences play important roles in the whole topic [116].

### 5.1. How Can ROS Support Carcinogenesis?

The first step of carcinogenesis is called initiation and refers to the alteration, change, or mutation of genes that develop either spontaneously or due to an exogenous source [17]. ROS can induce detrimental DNA damage via base modifications; inter-strand, intra-strand and DNA-protein crosslinks; and the induction of double strand breaks [151] (Figure 4). The promotion of cancer (cells) is considered a clonal expansion and accumulation of the pre-neoplastic cells that result from the initiation process, while progression already refers to a malignant conversion of the cells into invasive carcinoma [17,152]. Once tumors are initiated, there are several ways that ROS can drive promotion and expansion. These can be roughly divided into 3 subgroups: the impairment of transcription factors, signaling pathways, and epigenetics (Figure 4). We discuss the mechanisms most frequently involved in the majority of cancer subtypes but cannot acknowledge every mechanism in-depth.

The invasion of newly synthesized blood vessels to the network of tumor cells (angiogenesis) is an important event in carcinogenesis which is responsible not only for the supply with nutrients, immune cells, and oxygen but also for the disposal of waste products as well as tumor spreading (metastasis) [153] (Figure 4). One of the most prominent pro-angiogenic factors is the vascular endothelial growth factor (VEGF), which has been shown to be a key regulator in cancer angiogenesis upon stimulation via several pathways, factors, and conditions [154]. One of the transcription factors that leads to increased VEGF expression is HIF-1α [155]. Elevated ROS levels can suppress HIF-1α degradation, finally leading to an increased VEGF expression and subsequent angiogenesis in distinct cancer types such as prostate and ovarian cancer and fibrosarcoma [116,156]. Another important transcription factor in cancer is the nuclear factor kappa-light-chain-enhancer of activated B cells (NF-κB) [157]. Besides its well-known role in immune and inflammatory responses, cell proliferation, and apoptosis [158], NF-κB can also promote tumor proliferation [116,117]. Interestingly, mitochondrial ROS have been shown to activate NF-κB with the subsequent upregulation of the EGF receptor in pancreatic cancer, inducing the formation of pre-neoplastic lesions [117].

Another ROS-sensitive transcription factor involved in cell transformation, proliferation, and apoptosis is activator protein 1 (AP-1) [116,159]. In human colon cancer cells, the upregulation of AP-1 due to H_2_O_2_ has been documented, leading to increased MMP7 levels that are involved in tumor metastasis [160]. Furthermore, activated AP-1 enhances the expression of genes involved in growth stimulation like cyclin D1 but suppresses genes involved in the growth inhibition of cell cycle inhibitor p21, finally leading to an increased cell proliferation [116,161,162]. The tumor suppressor gene p53 is a key regulator of anti-proliferative cellular responses and is mutated in many tumor cells, resulting in a deleterious loss of function [163]. The ROS-dependent impairment of p53 expression and its activation/inactivation are controversially discussed in the literature. It is evident that ROS can directly inactivate p53 via the oxidation of cysteine residues in its DNA-binding domain [164], but the downstream effects of this inactivation remain ambivalent. In general, the inactivation of p53 will most likely leads to the loss of anti-proliferative effects, which are part of its tumor suppressor gene function. On the other hand, elevated ROS levels can promote apoptosis, senescence, and DNA-repair in a p53-dependent manner, finally leading to a reduced malignant transformation [116,165,166,167,168]. Of note, an impaired p53 expression can, in turn, have effects on ROS production itself, and functional p53 can enhance the expression of antioxidants like GPX, catalase, and SOD2 [116]. This downstream effect can be lost upon the mutation and absence of p53, finally leading to ROS accumulation and a pro-tumorigenic phenotype [116,169]. Again contradictorily, p53 has been shown to maintain mitochondrial health and to subsequently limit ROS generation and tumor development [116,170]. In summary, ROS-dependent p53 impairment and downstream effects are variable and seem to react in feedback loop systems that can be both pro- and anti-tumorigenic.

ROS can further influence other signaling pathways that are important for cancer progression such as the phosphatidylinositol-4,5-bisphosphate 3-kinase (PI3K) protein kinase B (Akt) pathway (PI3K/Akt) which plays an important role in cell metabolism, growth, proliferation, and survival [171]. Enhanced ROS levels have also been shown to inactivate negative regulators of this pathway (e.g., PTEN: phosphatase and tensin homolog and PTP1B: protein tyrosine phosphatase 1B) via cysteine oxidation [116,117,172,173].

Another possibility of how ROS can affect tumor development and progression is through epigenetic regulation. Epigenetic alterations in the TME are usually linked to DNA methylation or acetylation in the promoter regions of genes that are crucial for cancer cell proliferation or migration. In this regard, ROS can, for example, increase histone H3 acetylation of the promoter region of the Snai2 gene, which leads to an increased slug transcription factor expression and the subsequent enhancement of cell proliferation and migration [116,174]. Another study demonstrated a H_2_O_2_-dependent downregulation of E-cadherin in hepatocellular carcinoma cells that was described as a result of a hypermethylation in the promoter region upon the recruitment and ROS-dependent upregulation of histone deacetylase 1 (HDAC1) and DNA methyltransferase 1 (DNMT1). In turn, the loss of E-cadherin has been associated with epithelial-mesenchymal transition (EMT) that is favored by Slug and Snail expression, resulting in metastatic cancer [116,175,176]. Furthermore, DNA oxidation itself may lead to both hypermethylation and hypomethylation and subsequently inactivate tumor suppressor genes or activate oncogenes, respectively [116,177,178,179].

### 5.2. Sources of ROS in The Tumor Microenvironment

The importance of ROS for carcinogenesis immediately leads to the question of if there are ROS sources in the TME (Figure 4). Already 30 years ago, it was shown that cancer cells can induce pathologically increased ROS release [180]. However, analyzing ROS in the TME remains to be difficult, despite the fact that more and more techniques to measure different ROS are being developed (Table 1). A big challenge is that the unique TME is disrupted during preparation, leading to errors in analyses, in particular with regard to tumor cell metabolism. Furthermore, finding proper controls for in vitro studies is likewise challenging, as fast proliferating tumor cells can establish different ROS levels already due to their diverse metabolic state and not necessarily due to malignant transformation [181]. Nevertheless, several reliable studies have been performed to analyze ROS formation in the TME as reviewed in Reference [182]. The activation of oncogenes, the loss of tumor suppressor genes as well as mitochondrial DNA mutations and hypoxia may lead to an enhancement of ROS levels in tumor cells that further support carcinogenesis and malignancy [182]. Other publications further describe the same mechanisms to be responsible for the NOX-dependent ROS release by tumor cells [183,184].

As mentioned earlier, the cellular environment of a tumor includes several noncancerous cell types that are recruited upon tumor formation, such as cytokine-secreting T cells, macrophages, neutrophils, and fibroblasts. Cytokines are very important drivers of ROS production in the TME and, thus, further contribute to the mechanisms described in the section above. The exposure of tumor cells to certain important cytokines like IFNγ, TNFα, and IL-1 was shown to increase ROS production by tumor cells themselves in various cancer types [185], while the elevated ROS levels were attributed to NOX elevation or mitochondria activity [186]. In addition, neutrophils and macrophages are known to induce a rapid burst of superoxide formation during their killing activity, finally leading to the enhanced production of hydrogen peroxide in the TME [185,187,188].

In summary, several studies revealed the existence of both tumor cell-related and noncancerous sources of ROS that are available in a solid TME (Figure 4). Together, they build up a complex network, partially influencing each other and thereby further amplifying ROS formation in the TME. However, since each tumor has a distinct metabolic state, it is nearly impossible to make predictions on ROS levels in a given TME. The prediction of ROS levels is further complicated by the fact that the tumor is able to initiate an antioxidant defense upon a continuous exposure to high ROS concentrations.

### 5.3. What Are The Downstream Effects of Increased ROS in The Tumor Microenvironment?

Tumor-induced ROS release can further activate the signaling pathways discussed before, i.e., the modulation of transcription factors or epigenetic changes, which are involved in tumor initiation, promotion, and progression. This might induce a positive feedback loop system beneficial for tumor growth. In contrast, very high ROS levels may be toxic and may induce tumor cell death. To avoid this, tumor cells activate a defense mechanism to escape cell death [117]. This is of particular importance in metastatic tumors where cancer cells, detached from the extracellular matrix, are exposed to elevated ROS levels, for example, in oxidizing environments like blood and viscera [117,189,190]. Antioxidant pathways upregulated upon elevated ROS levels include nuclear factor erythroid 2 (Nrf2) upregulation, the activation of the JNK/p38 pathway, and GSH and NADPH elevation.

It is of great importance to distinguish between “good” and “bad” ROS with regard to cancer initiation and progression. In general, low ROS levels seem to be beneficial for tumor cells in order to support the proliferative and invasive properties, but upon crossing a distinct threshold, ROS can be toxic for tumor cells. Of note, some studies revealed that even low ROS levels can induce anti-tumorigenic signaling in cancer, leading, for example, to cell cycle arrest and senescence [116]. As an example, gene deletions in NOX4 have been documented in association with the hepatocellular carcinoma of a high tumor grade, and NOX4 knockdown studies showed an elevated proliferative capacity of liver tumor cells in vitro [191]. Several pathways are involved in anti-tumorigenic ROS-related signaling, finally leading to downstream tumor cell apoptosis, autophagy, or necroptosis [12].

Hence, tumor cells might exhibit an adaptive behavior in order to deal with different stages of ROS elevation. Apparently, tumor cells are able to distinguish between “good” and “bad” ROS and to subsequently induce either prooxidant or antioxidant mechanisms. Although various studies already proved antioxidant drugs to reduce cancer risk and progression [192], the application of this therapy needs to be reviewed intensely prior to prescription in order to ensure a clear benefit of the treatment.

## 6. Impact of ROS on STIM/Orai Channels

### 6.1. Impact of ROS on Orai

In addition to the ROS-TME interactions outline above, which have received a lot of attention in TME research, Orai channels are a relatively novel ROS target (Figure 5) but may play an important role in integrating Ca^2+^ and ROS signaling in the TME. ROS modulates the function of Orai channels, thereby modulating (Ca^2+^)_int_, which is of high relevance for tumor growth as discussed above. Bogeski et al. showed that endogenous and overexpressed Orai1 channels are inhibited by H_2_O_2_ with an IC_50_ of 34µM [15]. The same was found for HEK cells overexpressing Orai2. In contrast, HEK cells solely overexpressing Orai3 were not inhibited by H_2_O_2_ (Figure 5), indicating a ROS-sensitivity for Orai1 and Orai2 but not Orai3 [15].

Major targets of ROS are reactive cysteine residues [193]. Orai1 and Orai2 both contain three cysteine residues at amino acid positions 126, 143, and 195. Orai3 shares the first two cysteines but lacks the cysteine at position 195, which is replaced by a glycine. The substitution of Cys-195 with a serine in Orai1 conferred a partial resistance to H_2_O_2,_ whereas the insertion of a cysteine in Orai3 at the respective position rendered Orai3 H_2_O_2_ sensitive. Together, these experiments indicate a prominent role for Cys-195 in the redox-sensitivity of Orai channels [15]. If Orai channels were already coupled to and opened by a STIM protein, H_2_O_2_ did not block the Orai channels, indicating that STIM-Orai binding might mask Cys-195. This finding is particularly interesting for the STIM1 splice variant STIM1L primarily found in skeletal muscle cells [194]. In contrast to STIM1, STIM1L is coupled to Orai1 at resting Ca^2+^ concentrations without prior store-depletion which would leave STIM1L-Orai1 clusters insensitive to H_2_O_2_-mediated SOCE inhibition. The mode of action of the inhibition of Orai1 with H_2_O_2_ was attributed to an intramolecular interaction between the oxidized Cys-195, located in transmembrane helix 3 with a serine at amino acid position 239, located in transmembrane helix 4, causing the Orai channel to be in a closed conformation [14].

Considering the differences in redox-sensitivity between Orai1, Orai2, and Orai3 (Figure 5), the ratio between the isoforms might be an interesting factor regulating Ca^2+^ signals under oxidative stress conditions during pathophysiological situations like cancer. For primary human CD4^+^ T cells, it was indeed shown that naïve cells increase their Orai3 to Orai1 ratio when differentiating into effector cells, thereby reducing the channel’s redox-sensibility [15]. Since effector T cells are recruited to sites of inflammation, like the TME, where ROS concentrations are increased, increasing the Orai3 to Orai1 ratio could represent an adaptation mechanism to maintain Ca^2+^ signals necessary for proliferation and the production of cytokines. An Orai3 to Orai1 ratio change was also observed for monocytes in response to a bacterial peptide and following the bronchoalveolar lavage of S. aureus infected C57BL6/J mice [16]. Monocytes kill bacteria by rapid exocytosis of H_2_O_2_, which is produced by NOX2 in a SOCE-dependent manner [121,195]. Therefore, the switch to a less redox-sensitive composition of hexameric Orai channels would be beneficial to counter the H_2_O_2_-mediated inhibition of SOCE and a discontinuation of the killing pathway.

Not only immune cells but also cancer cells change the Orai3 to Orai1 ratio, thereby changing the H_2_O_2_ dependence of Orai-based Ca^2+^ entry. However, the findings are more complicated. Whereas in prostate cancer [196,197] and in basal breast cancer [115] the Orai3–Orai1 ratio was decreased, it was increased in another prostate cancer study [38], in estrogen receptor-positive breast cancer [65,115,198] and in non-basal breast cancer [115].

### 6.2. Impact of ROS on STIM

As discussed above, STIM1 and STIM2 differ in their sensitivity to luminal Ca^2+^ and their efficiency to gate Orai channels. STIM1 contains two cysteines at amino acid positions 49 and 56 which were reported to be redox sensitive [199,200] (Figure 5). Prins et al. [200] stated that under oxidizing conditions, a disulfide bond between Cys-49 and Cys-56 is formed. In contrast, Hawkins et al. [199] were not able to detect this disulfide bond formation but showed that the luminal Cys-56 is not oxidized by H_2_O_2_ but rather is S-glutathionylated under oxidizing conditions. This S-glutathionylation near the protein’s Ca^2+^ binding domain (formed by an EF-hand motive) rendered STIM1 constitutively active, thereby inducing SOCE independent of the Ca^2+^ filling state of the ER [199].

STIM2 is often described as the less abundant isoform because, in most tissues, STIM1 expression prevails [61,201,202]. However, in the brain, STIM2 is reported to be dominant, and neurons of STIM2-deficient mice are protected from apoptosis under oxidative stress, indicating a protective role for STIM2 [203].

## 7. Interactions between TME, Orai, and ROS: Promoting or Inhibiting Tumor Progression

Tumorigenesis usually starts by mutations of driver genes [204,205,206,207,208] such as RAS, BRAF, or NF-1. As tumors progress and the microenvironment develops, cells adapt to survive in this new environment, characterized by different levels of oxygen, nutrients, ROS, ATP, Ca^2+^, H^+^, cytokines, or waste products.

It is obvious from the analysis of publications in this review including references [5,10,52,79,86,209,210,211] that the expression of STIM-activated Orai channels is enhanced in most cancer types. In agreement with the finding that the amplitude of Ca^2+^ influx correlates very well with cell proliferation [212], it is expected that higher levels of STIM and/or Orai are beneficial for cancer growth and metastasis. On the other hand, it should not be neglected that there are no hints up to now that STIM and/or Orai channels work as cancer driver genes [5]. Nevertheless, the inhibition of Orai channels should inhibit cancer growth, and thus, the blockage of Orai1 and 2 by H_2_O_2_ [14,15,16] in the TME should inhibit cancer growth (see the Graphical abstract).

In several cancers, the upregulation of Orai3 was reported [38,39,65,66,67,68]. An increase of Orai3 would increase the Orai3–Orai1 ratio, thereby generating H_2_O_2_-insensitive heteromultimeric Orai channels (see the Graphical abstract). This, in turn, would allow enough Ca^2+^ entry in cancer cells to promote proliferation, metastasis, invasion, and survival. Thus, higher ROS levels in this case would be considered pro-tumorigenic, whereas in case the Orai3–Orai1 ratio is low and where mostly Orai1 homomultimers are formed, ROS levels would be considered anti-tumorigenic, resulting finally in cancer cell death. The dual role of ROS for cancer growth has indeed been reported [12] and many studies point to a beneficial role of a slightly oxidizing environment for tumor development, growth, and metastasis.

The efficient cytotoxicity of immune cells against cancer cells is a key factor to eradicating a tumor. Cytotoxic immune cells like cytotoxic T cells (CTL) or natural killer (NK) cells of the immune system also rely on Ca^2+^ influx through STIM-activated Orai channels to fulfill their cytotoxic functions [8,213,214,215,216]. Interestingly, a very low (Ca^2+^)_int_ is optimal for the high cytotoxic efficiency of CTL (and to a lesser amount also of NK cells) against cancer cells [216]. Thus, the inhibition of Ca^2+^ entry through Orai1 and Orai2 channels by slightly increased ROS in the TME may increase immune cell cytotoxicity against a tumor. In addition, CD4^+^ T cells and monocytes were found to increase their Orai3–Orai1 ratio upon activation [15,16]. This increase should protect immune cells against high ROS concentrations (in particular H_2_O_2_) in the TME to enable them to fulfill their cytotoxic functions (see the Graphical abstract). In general, the relative expression of STIMs and Orais and their respective ratios is of high importance for cell functions [41,42,43,44,45,46,47], and this should also be the case in cancer cells.

Another theory for the susceptibility of different cancers to ROS is the threshold concept, proposed two decades ago [217]. As a result of the elevated ROS concentrations in the TME, cancer cells are closer to the cytotoxic threshold of ROS than noncancerous cells. Consequently, the further application of ROS or the inhibition of antioxidants should lead to an imbalance in the interplay between ROS and the cell’s antioxidant system, leading to cell death or an increased sensitivity to additional therapeutics [217].

To better understand and predict the interplay between ROS and (Ca^2+^)_int_, local parallel measurements in the TME are necessary (see Table 1 for techniques to determine ROS levels) but not easy to achieve. For instance, the concentration dependence of ROS (and mainly of H_2_O_2_) regarding cancer cell proliferation or apoptosis and regarding CTL or NK cell cytotoxicity is unfortunately not well-defined. If different between cancer and immune cells, this would open up interesting therapeutic options. One pitfall to date is that, for many years, ROS measurements have relied on small chemical probes or cellular markers, which lack specificity and are prone to artifacts. It is now clear that redox changes are compartmentalized, pointing to the need of subcellular measurements and not only to rely on whole cell or even population measurements. Local effects and a lack of good redox probes are probably the reasons why there are many contradictions regarding the role of redox changes in cancer initiation, progression, metastasis, and therapy. First used as anticancer drugs [218,219], antioxidants have been shown to increase lung cancer progression [220] and melanoma metastasis in mice [221]. The same disappointing result occurred in the clinical trial SELECT, where dietary supplementation with vitamin E significantly increased the risk of prostate cancer among healthy men [222]. This clearly shows that, as expected, redox modulation in tumors is complex.

With the development of new methods using genetically encoded ratiometric redox sensors, it is now possible to identify ROS sources and different redox species within tumors even at subcellular levels [139,223,224]. The group of Dick et al. has expressed these sensors in a mouse model of non-small lung cancer xenograft and has observed different regions of mitochondrial redox state within the same tumor, varying from a very oxidized core, where a high percentage of necrotic cells was present to very reduced areas and regions of highly heterogeneous redox levels [223]. If most tumors are heterogeneous regarding their redox status as suggested by different O_2_, pH, and nutrient levels [3], the function of antioxidants (or oxidants) in the TME is not easy to predict.

## 8. Conclusions

STIM-activated Orai channels are likely important players in the TME to integrate Ca^2+^- and ROS-dependent functions in both cancer and immune cells (see the Graphical abstract). Considering the importance of local ROS and Ca^2+^ signaling in the TME, we need to better understand how/why these different tumor regions develop and, mainly, what their impact is on local cellular cancer and immune cell functions. Combining genetically encoded redox and Ca^2+^ sensors to analyze localized ROS concentrations and (Ca^2+^)_int_ in parallel in tumor tissues (if possible, in vivo) is important to test the relevance of ROS-modulated Orai channels in the TME in a quantitative manner and to allow reasonable predictions to target them pharmacologically.

## Figures and Tables

**Figure 1 cancers-11-00457-f001:**
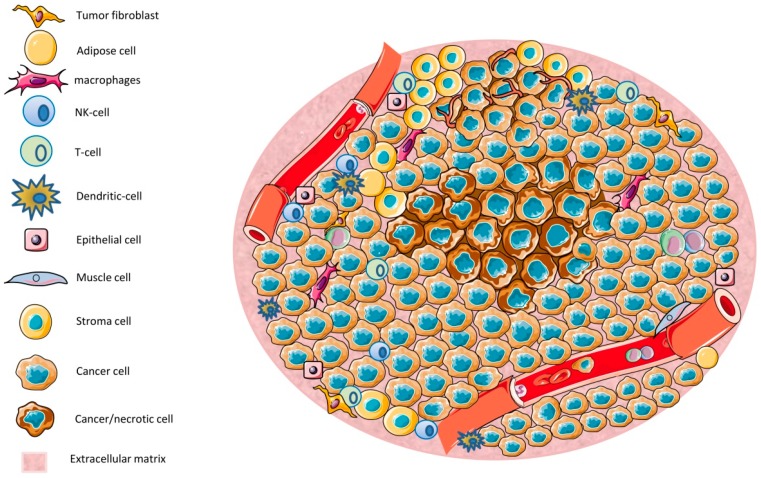
An overview of the tumor microenvironment (TME): The TME is composed by a diverse range of cell types, including tumor cells, immune cells, epithelial cells, and stromal cells. Areas of low nutrients and O_2_ result in necrotic regions. The TME controls tumor growth by diverse mechanisms that are further discussed in the text.

**Figure 2 cancers-11-00457-f002:**
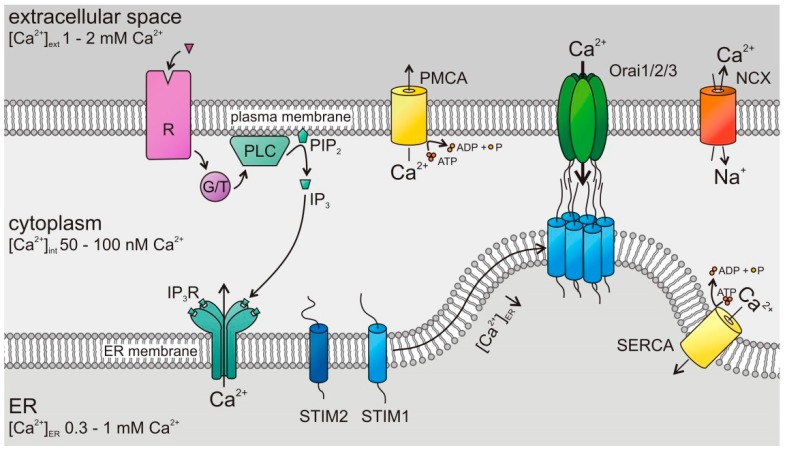
Store-operated Ca^2+^ entry (SOCE) by stromal-interaction molecule (STIM)/Orai channels in cancer cells: Following the stimulation of G protein- or tyrosine kinase-coupled (G/T) receptors (R), phospholipase C (PLC) hydrolyses phosphatidylinositol 4,5-bisphosphate (PIP_2_) to inositol trisphosphate (IP_3_). The latter binds to its receptor, a Ca^2+^ release channel in the endoplasmic reticulum (ER) membrane and opens it, which induces the Ca^2+^ depletion of ER Ca^2+^ stores. A drop of (Ca^2+^)_ER_ activates luminal Ca^2+^ sensor proteins, the STIMs. Activated STIMs oligomerize and move to the plasma membrane where they bind and open Orai channels, leading to a Ca^2+^ influx across the plasma membrane. Plasma membrane Ca^2+^ ATPases (PMCAs), Na^+^–Ca^2+^ exchangers (NCX), and sarco-/endoplasmic reticulum Ca^2+^ ATPases (SERCAs) export Ca^2+^ from the cytosol.

**Figure 3 cancers-11-00457-f003:**
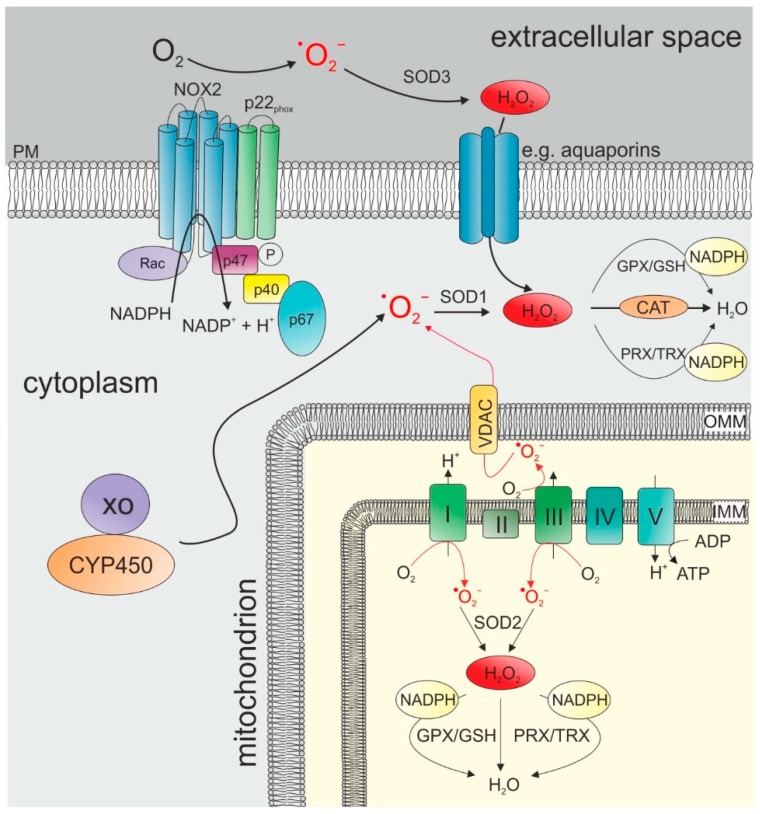
The main sources of ROS production and elimination: ROS can be generated either by activated NADPH oxidases (here represented by NOX2) located in the plasma membrane or by several complexes of the electron transport chain in the inner membrane of mitochondria. Other metabolic enzymes include xanthine oxidase (XO) and cytochrome P450 (CYP450) that directly form superoxide in the cytoplasm. Superoxide dismutases (SODs) can convert ^•^O_2_¯ to H_2_O_2_. SOD1 is located in the cytoplasm, SOD2 is located in the mitochondrial matrix, and SOD3 is located in the extracellular space. The further elimination of H_2_O_2_ via catalase, GPX/GSH, and/or PRX/TRX can either be initiated directly in the mitochondrial matrix or in the cytoplasm upon the transport of H_2_O_2_ via aquaporins. CYP450: cytochrome P450; IMM: inner mitochondrial membrane; OMM: outer mitochrondrial membrane; PM: plasma membrane; VDAC: voltage-dependent anion channel; XO: xanthine oxidase.

**Figure 4 cancers-11-00457-f004:**
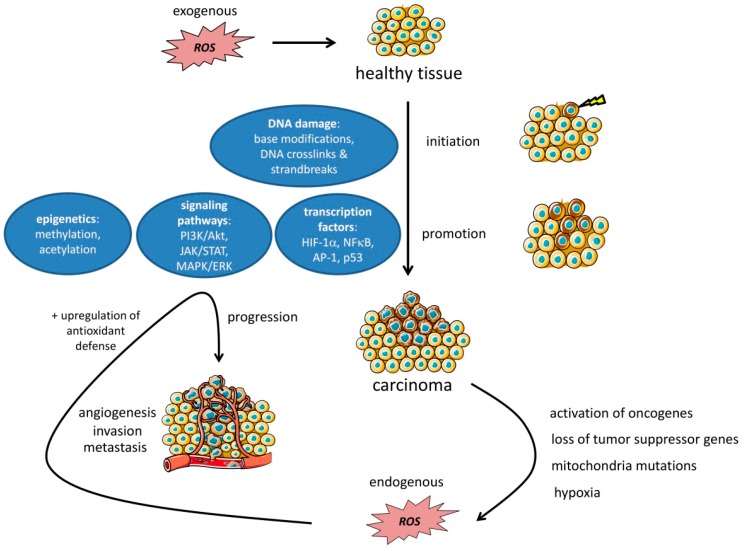
ROS in the TME: Non cancer-related ROS can be deleterious for healthy tissue and thereby lead to tumor initiation and promotion via DNA damage and the impairment of signaling pathways, transcription factor expression, and epigenetic changes. After the tumor initiation upregulation of oncogenes, the loss of tumor suppressor genes, mitochondria mutations, and hypoxic conditions can lead to further elevated tumor-related ROS levels. In a positive feedback loop system, rising ROS levels in the TME can in turn support tumor progression, angiogenesis, invasion, and metastasis via an amplification of the pathways involved in initiation and promotion. Finally, to protect against threshold-crossing toxic ROS levels, tumor cells initiate the upregulation of antioxidant defense mechanisms in order to prevent from ROS-related deleterious events like apoptosis or necroptosis.

**Figure 5 cancers-11-00457-f005:**
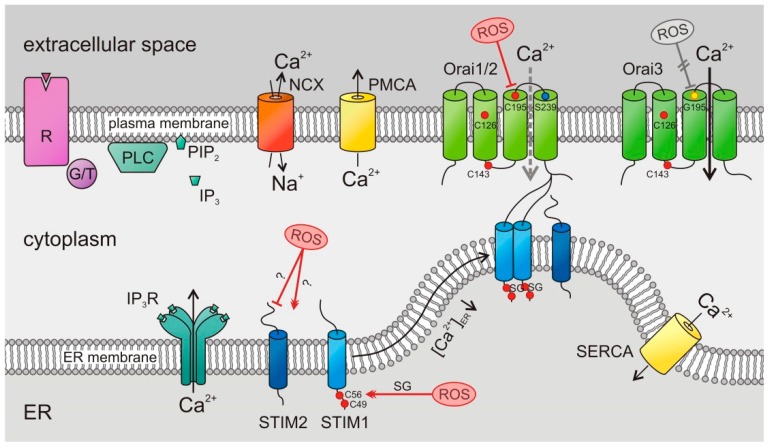
ROS interferes with STIM and Orai: Orai1 and Orai2 are blocked by ROS due to an oxidation of C195. Since Orai3 has no cysteine but a glycine on amino acid position 195 (G195), Orai3 channels are ROS-insensitive. ROS have an activating effect on STIM1. An S-glutathionylation (SG) of C56 renders STIM1 constitutively active. This has been reported to activate Orai channels without prior store-depletion (see text for details). R: receptor; G/T: G protein/tyrosine kinase; PLC: phospholipase C; PIP_2_: phosphatidylinositol 4,5-bisphosphate; IP_3_: inositol trisphosphate; ER: endoplasmic reticulum.

**Table 1 cancers-11-00457-t001:** A summary of the most commonly used tools and probes to measure various ROS.

Technique or Method	Tools and Examples	Specificity	(Potential) Applications
Fluorescence- based assays	Dihydroethidium (DHE)	^•^O_2_¯, if used with HPLC	in vitro, extra- and intracellular, cell suspensions
	Dihydrorhodamine (DHR)	not specific	
	2′,7′-dichlorodihydrofluorescein (DCFH_2_)	not specific	
	Amplex Red, Amplex UltraRed	H_2_O_2_	
	Hydroxyphenyl Fluorescein (HPF)	not specific	
	Aminophenyl Fluorescein (APF)	not specific	
Genetically encoded fluorescent probes	roGFP2	E_GSH_	in vivo and in vitro, intracellular, single cells, tissues, subcellular compartments
	roGFP2 coupled to glutaredoxins	E_GSH_	
	roGFP2 coupled to peroxidases or peroxiredoxins	H_2_O_2_	
	HyPer (different variants including HyPer-Red)	H_2_O_2_	
Chemiluminescence assays	Lucigenin	not specific	in vitro, cell suspensions
	Luminol		
	Isoluminol		
Enzymatic assays	Cytochrome C, Superoxide dismutase (SOD), Horseradish Peroxidase (HRP)	^•^O_2_¯	in vitro, cell suspensions
Chemical assays	Prussian Blue, Paraquat (1,1′-Dimethyl-4,4′bipyridium dichloride), FOX (containing xylenol orange)	H_2_O_2_, peroxides and others	in vitro, cell suspensions
Electrochemical assays	Electrodes of various types and sizes (macro-, mini-, micro- ultramicro- and nanoelectrodes), arrays, chips; additional redox mediators	^•^O_2_¯, H_2_O_2_ and other ROS	in vitro; single cells, cell suspensions, extra- and intracellular
Electron paramagnetic resonance (EPR) spectroscopy using spin probes	DMPO (5, 5-dimethyl-1-pyrroline-N-oxide)	^•^O_2_¯, OH^•^	in vitro and in vivo, cell suspensions, extra- and intracellular
	DEPMPO [5-(diethoxyphosphoryl)-5-methyl-1-pyrroline-N-oxide]	^•^O_2_¯, OH^•^	
	Horseradish Peroxidase assay (enzymatic using cyclic hydroxylamines)	H_2_O_2_	
	in vivo EPR with different functional spin traps and probes	Various ROS	

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
