# Peer review of "STIM-Orai Channels and Reactive Oxygen Species in the Tumor Microenvironment"

_cancers, 2019, doi:10.3390/cancers11040457_

Round 1
Reviewer 1 Report
This review by Frisch et al. provides a comprehensive and detailed critique of the interaction between ROS, Orai channels and calcium signals in the tumour microenvironment (TME). The authors nicely discuss the importance of the TME in cancer progression, the role of calcium signalling particularly store-operated calcium channels and the role of ROS. They then describe how ROS can influence Orai channel activity in an Orai homologue-specific manner. The review is timely, well-written and brings together different aspects of signalling in cancer. I have a few minor suggestions, which the authors might wish to consider.
1. Recent work by Monteith and colleagues (Cancers 2019) has shown that hypoxia can increase Orai3 expression in the MDA-MB-468 breast cancer cell line. Interestingly, despite the increase in channel expression, store-operated calcium entry was unaffected. This might point to a disconnect between tumour progression and calcium signalling, at least for Orai3 in this cancer. Regardless, the authors might like to discuss this in the section where they describe hypoxia and Orai1 (lines 320 onwards).
2. Given the understandable emphasis on ROS production in the TME, I think a Table briefly describing the different tools/probes used to measure ROS might be helpful for readers.
3. Line 144, perhaps plasma membrane NCX should also be mentioned as an effective means for extruding calcium from the cytosol.
Author Response
Reviewer#1: This review by Frisch et al. provides a comprehensive and detailed critique of the interaction between ROS, Orai channels and calcium signals in the tumour microenvironment (TME). The authors nicely discuss the importance of the TME in cancer progression, the role of calcium signalling particularly store-operated calcium channels and the role of ROS. They then describe how ROS can influence Orai channel activity in an Orai homologue-specific manner. The review is timely, well-written and brings together different aspects of signalling in cancer. I have a few minor suggestions, which the authors might wish to consider.
1. Recent work by Monteith and colleagues (Cancers 2019) has shown that hypoxia can increase Orai3 expression in the MDA-MB-468 breast cancer cell line. Interestingly, despite the increase in channel expression, store-operated calcium entry was unaffected. This might point to a disconnect between tumour progression and calcium signalling, at least for Orai3 in this cancer. Regardless, the authors might like to discuss this in the section where they describe hypoxia and Orai1 (lines 320 onwards).
Response: We have included the reference and discussed it as follows at the end of chapter 3.6, line 327: “Apart from the hypoxia-induced induction of Orai1 and STIMs, it was recently also reported that Orai3 expression was induced by HIF1α in MDA-MB-468 ERα negative cells {Azimi, 2019 #2984}. Unlike in ERα positive MCF-7 breast cancer cells {Faouzi, 2013 #1072;Motiani, 2013 #2073} where Orai3 silencing reduces SOCE, upregulation of Orai3 did not contribute to SOCE in ERα negative cells {Azimi, 2019 #2984}. However, since the Orai3 expression in breast cancer is highly dependent on ERα expression itself, one might speculate that the Orai1 complexes contribute significantly to Ca2+ signals in ERα negative cells. These results place the channels and sensors as new targets in regulating hypoxia.”
Reviewer#1: 2. Given the understandable emphasis on ROS production in the TME, I think a Table briefly describing the different tools/probes used to measure ROS might be helpful for readers.
Response: We agree and have included Table 1 (page 11, line 384). This table summarizes the most commonly used tools and probes to measure ROS. Regarding this comment we had to find a compromise between the comments by this reviewer and reviewer#2 who suggested to significantly shorten parts 4 and 5 of the ms. We agree with both reviewer#1 and #2 and have thus added the table but at the same time shortened parts 4 and 5.
Reviewer#1: 3. Line 144, perhaps plasma membrane NCX should also be mentioned as an effective means for extruding calcium from the cytosol.
Response: We agree and have included Na+-Ca2+ (NCX) exchangers on page 5 line 138. The new sentence now reads: “Plasma membrane Ca2+ ATPases (PMCAs), Na+-Ca2+ exchangers (NCX), and sarco-/endoplasmic reticulum Ca2+ ATPases (SERCAs) pump Ca2+ from the cytoplasm by converting adenosine triphosphate (ATP) to adenosine diphosphate (ADP) and phosphate (P).” NCX was also included in Figure 2.
Reviewer 2 Report
The review by Frisch et al covers an interesting topic regarding the function of ROS and calcium channels / signals in cancer. Chapter 6 is arguably the most unique aspect of this review where the authors try to synthesize the effects of ROS on Orai/STIM and the TME. Overall the review is worthwhile reading and provides a good summary of research conducted on Orai/STIM in cancer. The authors provide a somewhat encyclopedic overview without going into much detail about individual studies except for some by now older work on the ROS dependent regulation of Orais and STIM. Beyond a mere recitation of many studies, I would expect from this review a more critical review of the field that includes shortcomings of current studies, directions the field has to go into etc.
Many of the sections are composed of ‘one-sentence abstracts’ of various papers without any real narrative or critical assessment. As a reader one is a bit lost by the multitude of effects ROS and Ca2+ have on various aspects of cancer growth and progression, without any comments regarding the quality of the data underlying the claims made by the reviewed studies. In addition, I did not get a good sense which of the cited papers are based on experiments manipulating ORAI/STIM in cell lines and which have assessed this pathway in cancer models in vivo.
The overall impression a reader may get from this review is that ORAI/STIM are the next major drug target for cancer treatment. This is seems unlikely in my opinion because this pathway plays major physiological roles in many cell types and organs, which likely prevent it from being a good drug target in cancer. I am also not convinced based on the authors review, that there is enough good in vivo evidence for a major role of ORAi/STIM in cancer beyond cell line studies and correlative analyses of ORAI/STIM expression in cancer cells. Maybe the authors need to emphasize the crucial in vivo evidence in favor of a ORAI/STIM in cancer more clearly. To this point, I am not aware of efforts by drug companies to exploit ORAI/STIM inhibition for cancer treatment. If there are, this information could be added to the review.
Other comments:
The the effects of ROS on ORAI/STIM and the TME described in Chapter 6 are interesting, but it seems these effects are currently only shown in cells overexpressing ORAI/STIM genes, without validation in the context of cancer or immune cell function in vivo (at least the authors do not discuss such papers). This results, in Chapter 7, in an excessive discussion of Orai1/2/3 ratios and their potential effects on ROS sensitivity of tumor and immune cells. How much evidence is there really for such effects in the context cancer?
The review focuses on Orai/STIM/ROS mostly in cancer cells and relatively little effort is made to assess the role of Orai/STIM/ROS in antitumor immunity. Given the importance of Orai/STIM and ROS in immune function and the resurgence of cancer immunology research, this is a bit surprising, is there not more published on this topic?
Finally, I would recommend to shorten the review considerably. Chapter 4 on ROS production and elimination is a generic description of ROS that has been reviewed in this or similar form many times (same goes for Figure 3). Deleting or drastically shortening this chapter would provide a more focused review. The same might be said for Chapter 5, which deals with the effects of ROS on the TME. While this chapter is not a bad read, I am not sure how novel this section really is.
I would recommend to the authors to review the grammar and style. There are many grammatical errors and long sentences. Stylistically, it seems like the first half (Sections 1-3) and second half (6-8) are written by different authors, with a clear shift in style that is a bit difficult to adjust to. Overall, I think that the review could be more succinct, there is a lot of repetition.
Author Response
Reviewer#2: The review by Frisch et al covers an interesting topic regarding the function of ROS and calcium channels / signals in cancer. Chapter 6 is arguably the most unique aspect of this review where the authors try to synthesize the effects of ROS on Orai/STIM and the TME. Overall the review is worthwhile reading and provides a good summary of research conducted on Orai/STIM in cancer. The authors provide a somewhat encyclopedic overview without going into much detail about individual studies except for some by now older work on the ROS dependent regulation of Orais and STIM. Beyond a mere recitation of many studies, I would expect from this review a more critical review of the field that includes shortcomings of current studies, directions the field has to go into etc.
Many of the sections are composed of ‘one-sentence abstracts’ of various papers without any real narrative or critical assessment. As a reader one is a bit lost by the multitude of effects ROS and Ca2+ have on various aspects of cancer growth and progression, without any comments regarding the quality of the data underlying the claims made by the reviewed studies. In addition, I did not get a good sense which of the cited papers are based on experiments manipulating ORAI/STIM in cell lines and which have assessed this pathway in cancer models in vivo.
Response: We did not want to judge the quality of the papers in chapter 3. We have focused on the ones which we consider “more” solid than others. Since the publications cover many different mechansims/effects in many different cancer types, we did not find a better way to present them than the way we did.
We have tried to get a comprehensive list of citations which gives a very clear overview over the current state-of-the-art in this field. In the following chapters we have put this work in perspective.
We agree that it makes sense to stress the in vivo publications and we have done that now in the following parts:
Chapter 3.3
“SOCE alters cancer cell proliferation in vitro {Wang, 2015 #2130;Kim, 2014 #2203;Zhou, 2017 #2192} and also in vivo {Chen, 2011 #2079;Zhu, 2014 #2134;Gui, 2016 #2183;Li, 2013 #2204;Xia, 2016 #2123;Li, 2017 #2270}. However, how Ca2+ controls distinct checkpoints is not well understood. Increases in basal or transient fluctuation in Ca2+ are involved but also [Ca2+]ext needs to be considered. Cell cycle arrest in G0/G1 phase in U251 cells {Li, 2013 #2204}, neck squamous cell carcinoma cell lines {Li, 2017 #2270} and at the S and G2/M phases in cervical cancer cells {Chen, 2011 #2079} by STIM1-silencing has been reported. A pro-proliferative role of STIM1 in vivo using U251 human glioma xenograft model in mice revealed that knocking-down STIM1 in xenografts demonstrated diminished growth {Li, 2013 #2204}.”
“Pharmacological inhibition or knocking-down of Orai channel could block human esophageal squamous cell carcinoma proliferation in vitro and tumor growth in vivo {Zhu, 2014 #2134}.”
“Overexpression of STIM2 inhibits cell proliferation and tumor growth in colorectal cancers in vivo {Aytes, 2012 #2138} but promotes cell migration in primary melanoma in vivo {Stanisz, 2014 #2141}, implicating the contribution of STIM2 signaling at different stages of tumor progression. Furthermore, the high Orai1 and STIM2 expression found in melanoma biopsies at the rim of invading tumors, are linking their possible role in tumor invasion and/or metastasis in vivo {Stanisz, 2014 #2141}.”
“By using mice xenograft models, it was shown that Orai3 plays a crucial role in prostate cancer development in vivo {Dubois, 2014 #2090}.”
“Silencing of ERα causes decrease expression of Orai3 and cell proliferation in vitro {Motiani, 2013 #2073}. The same study places Orai3 as an important player in tumorigenesis in vivo, since the growth of breast tumors was significantly reduced by Orai3 knock-down before transfer to recipient SCID mice {Motiani, 2013 #2073}.”
Chapter 3.6
“Over the last ten years, in vitro and in vivo evidence has accumulated that SOCE components are involved in cell motility, invasion and tumor metastasis {Mo, 2018 #2850}.”
Reviewer#2: The overall impression a reader may get from this review is that ORAI/STIM are the next major drug target for cancer treatment. This is seems unlikely in my opinion because this pathway plays major physiological roles in many cell types and organs, which likely prevent it from being a good drug target in cancer. I am also not convinced based on the authors review, that there is enough good in vivo evidence for a major role of ORAi/STIM in cancer beyond cell line studies and correlative analyses of ORAI/STIM expression in cancer cells. Maybe the authors need to emphasize the crucial in vivo evidence in favor of a ORAI/STIM in cancer more clearly. To this point, I am not aware of efforts by drug companies to exploit ORAI/STIM inhibition for cancer treatment. If there are, this information could be added to the review.
We agree with the rewiewer that we have overstated “ORAI/STIM as the next major drug target for cancer treatment”. This idea is put forward by many papers and reviews in the Orai field. As the reviewer suggested we have highlighted the in vivo publications (see previous response).
Nevertheless, we have weakened the therapeutic potential of Orai channels in several sections including:
Line 41: “Since malignant cells exhibit a strong dependence on Ca2+ flux for proliferation, Orai channels could be considered a potential therapeutic target to inhibit cancer growth”.
Line 52: “In this review, we focus on interactions of Orai channels and ROS in the TME and their potential relevance for TME development.”
Line 117: “Since malignant cells functions depend on Ca2+ flux, considerable interest has emerged in the therapeutic potential of inhibiting Orai for many cancer types.”
Nevertheless, considering the redox dependence of Orai channels and the dependence of cellular functions on different calcium signals (in space, amplitude and time), Orai channels appear to be an interesting target to change cancer cell fate.
Reviewer#2: Other comments:
The the effects of ROS on ORAI/STIM and the TME described in Chapter 6 are interesting, but it seems these effects are currently only shown in cells overexpressing ORAI/STIM genes, without validation in the context of cancer or immune cell function in vivo (at least the authors do not discuss such papers). This results, in Chapter 7, in an excessive discussion of Orai1/2/3 ratios and their potential effects on ROS sensitivity of tumor and immune cells. How much evidence is there really for such effects in the context cancer?
We agree with the reviewer that “the effects of ROS on ORAI/STIM and the TME described in Chapter 6 are interesting”. The reviewer states that the redox effect on Orai channels is only described in over-expressed cells. Unfortunately, we did not make it clear that endogenous Orais are inhibited by ROS. In the initial version of the ms, in chapter 6 we only discussed the data in cells overexpressing ORAI/STIM genes as the reviewer states. In chapter 7 we then also included the data on endogenous Orais in immune cells. We apologize for not presenting this more clearly. In the first paper (Bogeski et al Science Signaling 2010), redox regulation is also shown for endogenous Orai in immune cells. We have now added this information also in chapter 6 to avoid confusion. This part reads (line 544): “Bogeski et al. showed that endogenous and over-expressed Orai1 channels are inhibited by H2O2 with an IC50 of 34µM”.
There is to our knowledge no in vivo data in the whole animal. Nevertheless, as the reviewer stated this mechanism is interesting and since it is probably relevant in cancer cells and in immune cells, we have discussed it in detail in chapter 7. This mechanism within the main focus of the review.
Reviewer#2: The review focuses on Orai/STIM/ROS mostly in cancer cells and relatively little effort is made to assess the role of Orai/STIM/ROS in antitumor immunity. Given the importance of Orai/STIM and ROS in immune function and the resurgence of cancer immunology research, this is a bit surprising, is there not more published on this topic?
Response: We did not aim to address Orai/STIM/ROS in immune cells and antitumor immunity apart from two exceptions (see below). We agree with the reviewer that this would be an interesting topic to discuss, albeit in our opinion still quite controversial. To our knowledge there are at least two possible mechanisms how Orai channels and SOCE are involved in antitumor immunity. 1) Activation of calcium-dep. transcription factors, one of the main calcium-dep. functions in immune cells, 2) Calcium dep. production and release of cytokines, chemokines and cytotoxic substances (perforin, granzymes etc.) to eliminate tumor cells. We have included the second part into chapter 7 because there is very good evidence of this particular Orai function in cytotoxic immune cells. Another part we included, is the redox regulation of Orai channels because it is well established and may be relevant to antitumor immunity.
Reviewer#2: Finally, I would recommend to shorten the review considerably. Chapter 4 on ROS production and elimination is a generic description of ROS that has been reviewed in this or similar form many times (same goes for Figure 3). Deleting or drastically shortening this chapter would provide a more focused review. The same might be said for Chapter 5, which deals with the effects of ROS on the TME. While this chapter is not a bad read, I am not sure how novel this section really is.
Response: This reviewer suggests to significantly shorten chapter 4 and 5. On the other hand reviewer#1 wants more detail because she/he states that the emphasis on ROS in understandable (“Given the understandable emphasis on ROS production in the TME”) and she/he even suggests to add a table summarizing the technologies used to quantify ROS.
In a way, we agree with both reviewers. Thus we have significantly shortened both chapters 4 and 5 (by about 1500 words). We believe that these parts are more clear now but still emphazise the important informations. Considering reviewer#1, we have added Table 1 to summarize commonly used technologies to measure ROS.
Reviewer#2: I would recommend to the authors to review the grammar and style. There are many grammatical errors and long sentences. Stylistically, it seems like the first half (Sections 1-3) and second half (6-8) are written by different authors, with a clear shift in style that is a bit difficult to adjust to. Overall, I think that the review could be more succinct, there is a lot of repetition.
Response: Of course, as the reviewer stated, different authors have contributed initial drafts of different parts. However, the distribution is not as the reviewer thought it is. For instance, initial drafts of chapters 1, 2 and 7, 8 are written by the same author. All authors have polished all parts.
As the reviewer suggested, we have improved style, grammar etc in the whole ms. Changes are marked. However, we also want to mention that the other two reviewers do not have a problem with style, grammar etc.
Reviewer 3 Report
March 12, 2019
In this manuscript, Frisch et al. summarize the current knowledge about the STIM/Orai Ca2+ channels and reactive oxygen species (ROS) in the tumor microenvironment. These two effectors have been explored deeply during the process of cell apoptosis. More recently, the Ca2+ and ROS transients in the tumor microenvironment appear to play a major role in a vast variety of pro-survival signaling pathways that may be crucial for the fate of tumors. The authors have seen this big picture, so proposed a scenario that redox changes alter Ca2+ entry through Orai channels in both malignant and non-malignant cells, such as immune cells, resulting in changes in [Ca2+]int with direct impact on tumor fate. The questions raised in the end of this review are potentially interesting to the field.
I only have several recommendations and one minor typo:
Recommendations:
1. Please consider using the title of “STIM/Orai Ca2+ channels and reactive oxygen species in the tumor microenvironment” instead of the current title.
2. On page 4, in Figure 2, please consider using other way to express multimeric STIM to avoid misleading readers. Because there is no data to support this stoichiometry of 3 STIM1s and 3 STIM2s during the activation stage of Orai channels.
3. On page 4, in Figure 2, please consider using a tetramer to redraw IP3R. Do the same on page 17 for Figure 5.
Minor typo:
In line 35, ion concentrations should be “ions”. Do the same change in line 82.
Author Response
Reviewer#3: In this manuscript, Frisch et al. summarize the current knowledge about the STIM/Orai Ca2+ channels and reactive oxygen species (ROS) in the tumor microenvironment. These two effectors have been explored deeply during the process of cell apoptosis. More recently, the Ca2+ and ROS transients in the tumor microenvironment appear to play a major role in a vast variety of pro-survival signaling pathways that may be crucial for the fate of tumors. The authors have seen this big picture, so proposed a scenario that redox changes alter Ca2+ entry through Orai channels in both malignant and non-malignant cells, such as immune cells, resulting in changes in [Ca2+]int with direct impact on tumor fate. The questions raised in the end of this review are potentially interesting to the field.
I only have several recommendations and one minor typo:
Recommendations:
Reviewer#3: 1. Please consider using the title of “STIM/Orai Ca2+ channels and reactive oxygen species in the tumor microenvironment” instead of the current title.
Response: We agree with the reviewer and have changed the title.
Reviewer#3: 2. On page 4, in Figure 2, please consider using other way to express multimeric STIM to avoid misleading readers. Because there is no data to support this stoichiometry of 3 STIM1s and 3 STIM2s during the activation stage of Orai channels.
Response: We agree and have changed Figure 2 and also the Graphical abstract. We now show a homomultimer. In the text we discuss that the stoichiometry is a matter of debate. The text reads: “The two known SOCE activators, STIM1 and STIM2, sense the ER Ca2+ store content and activate Orai channels in the plasma membrane. They are known to form homomultimers (as depicted in Figure 2) but may also form heteromultimers” (ref to: Subedi KP, Ong HL, Son GY, Liu X, Ambudkar IS. STIM2 Induces Activated Conformation of STIM1 to Control Orai1 Function in ER-PM Junctions. Cell Rep. 2018 Apr 10;23(2):522-534. doi: 10.1016/j.celrep.2018.03.065. PMID: 29642009).
Reviewer#3: 3. On page 4, in Figure 2, please consider using a tetramer to redraw IP3R. Do the same on page 17 for Figure 5.
Response: We agree and have changed Figs 2. and 5. IP3R is now shown as a tetramer.
Reviewer#3: Minor typo:
In line 35, ion concentrations should be “ions”. Do the same change in line 82.
Response: We agree and have replaced “ion concentrations” by “ions”.